# Recovering Plasticity of Neural Networks via Soft Weight Rescaling

## Abstract

Recent studies have shown that as training progresses, neural networks gradually lose their capacity to learn new information, a phenomenon known as plasticity loss. An unbounded weight growth is one of the main causes of plasticity loss. Furthermore, it harms generalization capability and disrupts optimization dynamics. Re-initializing the network can be a solution, but it results in the loss of learned information, leading to performance drops. In this paper, we propose Soft Weight Rescaling (SWR), a novel approach that prevents unbounded weight growth without losing information. SWR recovers the plasticity of the network by simply scaling down the weight at each step of the learning process. We theoretically prove that SWR bounds weight magnitude and balances weight magnitude between layers. Our experiment shows that SWR improves performance on warm-start learning, continual learning, and single-task learning setups on standard image classification benchmarks.

## 1 Introduction

Recent works have revealed that a neural network loses its ability to learn new data as training progresses, a phenomenon known as plasticity loss. A pre-trained neural network shows inferior performance compared to a newly initialized model when trained on the same data (Ash & Adams, 2020; Berariu et al., 2021). Lyle et al. (2024b) demonstrated that unbounded weight growth is one of the main causes of plasticity loss and suggested weight decay and layer normalization as solutions. Several recent studies on plasticity loss have proposed weight regularization methods to address this issue (Kumar et al., 2023; Lewandowski et al., 2023; Elsayed et al., 2024). Unbounded weight growth is a consistent problem in the field of deep learning; it is problematic not only for plasticity loss but also undermines the generalization ability of neural networks (Golowich et al., 2018; Zhang et al., 2021) and their robustness to distribution shifts. Increasing model sensitivity, where a small change in the model input leads to a large change in the model output, is also closely related to the magnitude of the weights. Therefore, weight regularization methods are widely used in various areas of deep learning and have been consistently studied.

Weight regularization methods have been proposed in various forms, including additional loss terms (Krogh & Hertz, 1991; Kumar et al., 2023) and re-initialization strategies (Ash & Adams, 2020; Li et al., 2020b; Taha et al., 2021). The former approach adds an extra loss term to the objective function, which regularizes the weights of the model. These approaches are used not only to penalize large weights but also for other purposes, such as knowledge distillation (Shen et al., 2024). However, they can cause optimization difficulties or conflict with the main learning objective, making it harder for the model to converge effectively (Ghiasi et al., 2024). Liu et al. (2021) also proved that the norm penalty of a family of weight regularizations weakens as the network depth increases. Moreover, such methods require additional gradient computations, resulting in slower training. In addition, several studies argued that regularization methods could be problematic with normalization layers. For instance, weight decay destabilizes optimization in weight normalization (Li et al., 2020a), and interferes learning with batch normalization (Lyle et al., 2024b), both of which can hinder convergence. On the other hand, re-initialization methods are aimed at resetting certain parameters of the model during training to escape poor local minima and encourage better exploration of the loss landscape. Zaidi et al. (2023) demonstrated that re-initialization methods improve generalization even with modern training protocols. While re-initialization methods improve generalization ability, they raise the problem of losing knowledge from previously learned data (Zaidi et al., 2023;

Ramkumar et al., 2023; Lee et al., 2024; Shin et al., 2024). It leads to a notable performance drop, especially problematic when access to the previous data is unavailable.

In this paper, we propose a novel weight regularization method that has advantages of both of those two approaches. Our method, Soft Weight Rescaling (SWR), directly reduces the weight magnitudes close to the initial values by scaling down weights. With a minimal computational overhead, it effectively prevents unbounded weight growth. Unlike previous methods, SWR recovers plasticity without losing information. In addition, our theoretical analysis proves that SWR bounds weight magnitude and balances weight magnitude between layers. We evaluate the effectiveness of SWR on standard image classification benchmarks across various scenarios—including warm-start learning, continual learning, and single-task learning—comparing it with other regularization methods and highlighting its advantages, particularly in the case of VGG-16.

The contributions of this work are summarized as follows. First, We introduce a novel method that effectively prevents unbounded weight growth while preserving previously learned information and maintaining network plasticity. Second, we provide a theoretical analysis demonstrating that SWR bounds the magnitude of the weights and balances the weight magnitude across layers without degrading model performance. Finally, we empirically show that SWR improves generalization performance across various learning scenarios.

The rest of this paper is organized as follows. Section 2 reviews studies on weight magnitude and regularization methods. In Section 3, we explain weight rescaling and propose a novel regularization method, Soft Weight Rescaling. Then, in Section 4, we evaluate the effectiveness of Soft Weight Rescaling by comparing it with other regularization methods across various experimental settings.

## 2 RELATED WORKS

**Unbounded Weight Growth.** There have been studies associated with the weight magnitude. Krogh & Hertz (1991); Bartlett (1996) indicated that the magnitude of weights is related to generalization performance. Besides, as the magnitude of the weights increases, the Lipschitz constant also tends to grow (Couellan, 2021). This leads to higher sensitivity of the network, potentially affecting its stability and generalization. Ghiasi et al. (2024) demonstrated that weight decay plays a role in reducing sensitivity for noise. Moreover, Lyle et al. (2024b) claimed that unbounded weight growth is one of the factors of plasticity loss in training with non-stationary distribution. These studies indicate that enormous weight magnitudes disturb effective learning. Unfortunately, weight growth is inevitable in deep learning. Neyshabur et al. (2017) showed that when the training error converges to 0, the weight magnitude gets unbounded. Merrill et al. (2020) observed that weight magnitude increases with $O(\sqrt{t})$, where $t$ is the update step during transformer training. These explanations highlight the ongoing need for weight regularization in modern deep learning.

**Weight Regularization.** Various methods have been proposed to regularize the weight magnitude. L2 regularization, which is also termed as weight decay, is a method to apply an additional loss term that penalizes the L2 norm of weight. Although it is a method widely used, several studies pointed out its problems (Ishii & Sato, 2018; Liu et al., 2021). Yoshida & Miyato (2017) suggested regularizing the spectral norm of the weight matrix and showed improved generalization performance in various experiments. Kumar et al. (2020) regularized the weights to maintain the effective rank of the features. On the other hand, several studies have explored how to utilize the initialized weights. Kumar et al. (2023) imposed a penalty on L2 distance from initial weight and Lewandowski et al. (2023) proposed using the empirical Wasserstein distance to prevent deviating from initial distribution. However, these methods require additional gradient computations.

**Re-initialization methods.** Ash & Adams (2020) demonstrated that a pre-trained neural network achieves reduced generalization performance compared to a newly initialized model. The naive solution is to initialize models and train again from scratch whenever new data is added, which is very inefficient. Based on the idea that higher layers learn task-specific knowledge, methods that re-initialize the model layer by layer, such as resetting the fully-connected layers only (Li et al., 2020b), have been proposed. To explore a more efficient approach, several attempts have been made to re-initialize the subnetwork of the model (Han et al., 2016; Taha et al., 2021; Ramkumar et al., 2023; Sokar et al., 2023). In particular, Ramkumar et al. (2023) calculated the weight importance and re-initialized the task-irrelevant parameters. Sokar et al. (2023) proposed to reset dormant nodes

which do not influence the model. However, these methods pose a new drawback in additional computational cost. On the other hand, there have been presented weight rescaling methods that leverage initial weight. Alabdulmohsin et al. (2021) proposed the Layerwise method which rescales the first $t$ blocks to have their initial norms and re-initializes all layers after $t$-th layer, for the training stage $t$. More recently, Niehaus et al. (2024) introduced the Weight Rescaling method, which rescales weight to enforce the standard deviation of weight to initialization. The limitation of these two weight rescaling methods is that they depend on the model architecture and require to find a proper rescaling interval.

## 3 METHOD

In this section, we introduce the proportionality of neural networks to explain a weight regularizing method that preserves the behavior of the model. Next, we demonstrate that our method, SWR, regularizes learnable parameters while satisfying the property. Finally, we will discuss the reason for the importance of the proportionality and advantage of SWR that improves model balancedness.

### 3.1 NOTATIONS

Let $f_\theta$ be a neural network with $L$ layers and activation function $\phi$, where the input $x \in \mathbb{R}^m$ and the output $z \in \mathbb{R}^n$. The set of learnable parameters is denoted by $\theta$, comprising the weight matrices $W_l$ and bias vectors $b_l$ of the $l$-th layer. Let $a_l$ represent the vector of activation outputs of the $l$-th layer, and $z_l$ the pre-activation outputs before applying the activation function. The final output of the network $z = f_\theta(x)$ is obtained recursively as follows:

$$a_0 \doteq x$$
$$z_i = W_i a_{i-1} + b_i, \quad i \in \{1, ..., L-1\}$$
$$a_i = \phi(z_i), \quad i \in \{1, ..., L-1\}$$
$$z = W_L a_{L-1} + b_L,$$

where $z_L = z$.

For convenience, the norm expression of a matrix will be considered an element-wise L2 norm, which is known as the Frobenius norm: $\|W\| \doteq \|W\|_F = \sqrt{\sum_i \sum_j |w_{ij}|^2}$, where $w_{ij}$ represents an element of the matrix $W$. Additionally, we consider multiplying a constant by a matrix or vector as element-wise multiplication.

### 3.2 WEIGHT RESCALING

Previous studies have suggested regularizing the magnitude or spectral norm by multiplying the parameters by a specific constant (Huang et al., 2017; Ash & Adams, 2020; Gogianu et al., 2021; Gouk et al., 2021; Niehaus et al., 2024). However, rescaling the weights can alter the behavior of models, except in specific cases (e.g. a neural network without biases). It is clear that when a constant is multiplied by the weight matrix and bias of the final layer, the network output will be scaled accordingly. However, it becomes complicated when the scaling constant varies across layers. To resolve this complexity, we demonstrate in Theorem 1 that it is possible to avoid decreasing the model's accuracy by employing a specific scaling method. We will first outline the relevant properties in the form of Definition 1.

**Definition 1** (*Proportionality of neural network*). *Let the neural network $f_{\theta'}$ have the same input and output dimension with $f_\theta$. Then, we say that $f_{\theta'}$ and $f_\theta$ are proportional if and only if*

$$f_{\theta'}(x) = k \cdot f_\theta(x)$$

*for a real constant $k$ and all input data $x$. We refer to the constant $k$ as the proportionality constant of $f_\theta$ and $f_{\theta'}$.*

We investigated the following theorem shows that it is always possible to construct a proportional network for any arbitrary neural network.

**Theorem 1.** *Let $f_\theta$ be a feed-forward neural network with affine, convolution layers, and homogeneous activation functions (e.g. ReLU, Leaky ReLU, etc.). For any positive real number $C$, we can find infinitely many networks that are proportional to $f_\theta$ with proportionality constant $C$.*

We will briefly explain how to find the network that is proportional to $f_\theta$. Let a network that has $L$ layers be $f_\theta$, and a set $c = \{c_1, c_2, \ldots, c_L\}$ consisting of positive real numbers such that $C = \Pi_{i=1}^L c_i$. Then, construct the new parameter set $\theta^c \doteq \{W_1^c, b_1^c, \ldots W_L^c, b_L^c\}$ by rescaling parameters with the following rules:

$$W_l^c \leftarrow c_l \cdot W_l, \quad b_l^c \leftarrow \left(\prod_{i=1}^l c_i\right) \cdot b_l$$

Then, for all input $x$, it satisfies $f_{\theta^c}(x) = C f_\theta(x)$. A detailed proof can be found in Appendix A.

In the following, scaled network $f_{\theta^c}$, final cumulative scaler $C$, and the scaler set $c$ will refer to the definitions provided above. Note that Theorem 1 indicates that two proportional neural networks have identical behavior in classification tasks. This suggests that scaling the bias vectors according to a certain rule allows for regularization without affecting the model's performance. It remains the same for the case of any homogeneous layer, such as max-pooling or average-pooling.

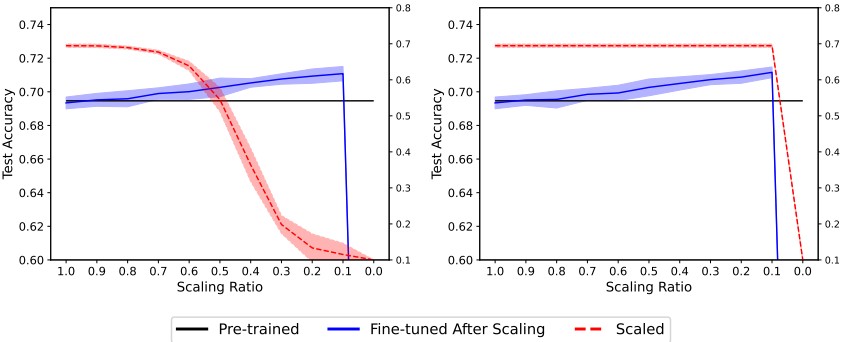

Figure 1: **An illustrative comparison of the proportionality.** The left figure shows the results of weight scaling without considering proportionality, while the right figure shows the results when proportionality is accounted for. The dashed line represents the test accuracy right after scaling, and the solid lines represent the best test accuracy achieved through additional training. All results are averaged over 5 runs on the CIFAR-10 dataset.

An example illustrating the effect of the proportionality is shown in Fig. 1. The left figure represents the outcomes of weight scaling without taking proportionality into account, and the right represents the results when proportionality is considered. Two scaling approaches are compared across different scaling magnitudes on the CIFAR-10 dataset (Krizhevsky et al., 2009). The black horizontal line denotes the best test accuracy achieved during training over 100 epochs, and the blue line represents the best test accuracy during an additional 50 epochs of training. All scaling methods outperformed the best accuracy of the pre-trained model (black), indicating that the scaling method can address the overfitting issue. However, it is notable that considering proportionality as Theorem 1 maintains its test accuracy perfectly across all scaling ratios, as indicated by the red line. In contrast, the performance of the opposite exhibits a decline as the scaling magnitude increases. However, as mentioned above, there are infinitely many ways to rescale parameters. In the following section, we will discuss how to determine the scaler set $c$.

### 3.3 SOFT WEIGHT RESCALING

Selecting different scaling factors per layer becomes impractical as the number of layers increases. In this subsection, we propose a novel method for effectively scaling parameters; the scaling factor of each layer depends on the change rate of the layer. We define the rate of how much the model has changed from the initial state as the ratio between the Frobenius norm of the current weight matrix and that of the initial one. Therefore, the scaling factor of the $l$-th layer is $c_l = \|W_l^{\text{init}}\|/\|W_l\|$. This ensures that the magnitude of the layer remains at the initial value, and may constrain the model,

forcing the weight norm to remain unchanged from the initial magnitude. Since the initial weight norm is small in most initialization techniques, the model may lack sufficient complexity (Neyshabur et al., 2015b). To address this limitation, we alleviate the scaling factor as follows:

$$c_l = \frac{\lambda \times \|W_l^{\text{init}}\| + (1 - \lambda) \times \|W_l\|}{\|W_l\|}$$

With an exponential moving average (EMA), models can deviate from initialization smoothly while still regularizing the model. While this modification breaks hard constraints for weight magnitude, the algorithm still prevents unlimited growth of weight. We presented the proof of the boundedness of the weight magnitude in Appendix B.

It is natural to question whether Theorem 1 can also be applied to networks that utilize commonly used techniques such as batch normalization (Ioffe, 2015) or layer normalization (Ba, 2016), due to their scale-invariant property (which is, if $g$ is a function of normalization layer, for input $x$, $g(cx) = g(x)$ for $\forall c > 0$). However, this property implies that we only need to focus on the learnable parameters of the final normalization layer to maintain the proportionality. The algorithm, including the normalization layer, is provided in Algorithm 1. For simplicity, we denote the scale and shift parameters of the normalization layer as $W$ and $b$ just like a typical layer, and in the case of layers without a bias vector (e.g. like the convolution layer right before batch normalization), we consider bias as the zero constant vector.

---

**Algorithm 1** Soft Weight Rescaling

**Given:** Data stream $\mathcal{D}$, neural network $f_\theta$ with learnable parameters $\{(W_1, b_1), \ldots, (W_L, b_L)\}$.
**Initialize:** step size $\alpha$, coefficient $\lambda$
$n_l^{\text{init}} \leftarrow \|W_l\|, l \in \{1, \ldots, L\}$
$k \leftarrow \begin{cases} \text{Index of final normalization layer,} & \text{if network has normalization layer} \\ 0, & \text{otherwise} \end{cases}$
**for** $(x, y)$ in $\mathcal{D}$ **do**
$\quad \theta \leftarrow$ Parameters after **Gradient update** for $(x, y)$ $\quad \triangleright$ e.g. update with CrossEntropyLoss
$\quad C \leftarrow 1$ $\quad\quad\quad\quad\quad\quad\quad\quad\quad\quad\quad\quad\quad\quad\quad\quad\quad\quad\quad \triangleright$ variable to calculate cumulative scaler
$\quad$ **for** $l$ in $\{1, 2, \ldots, L\}$ **do**
$\quad\quad c_l \leftarrow \frac{\lambda n_l^{\text{init}} + (1 - \lambda)\|W_l\|}{\|W_l\|}$
$\quad\quad C \leftarrow \begin{cases} c_l \cdot C & \text{if } l \geq k \\ c_l & \text{otherwise} \end{cases}$ $\quad\quad\quad \triangleright$ cumulate scalers from last normalization layer
$\quad\quad (W_l, b_l) \leftarrow (c_l \cdot W_l, C \cdot b_l)$
$\quad$ **end for**
**end for**

---

It is notable that SWR scales the weights preceding the final normalization layer, while they do not affect the scale of the output. However, each of them has a distinct role. First, for convolution layers, the scalers control the effective learning rate which has been studied in previous research (Van Laarhoven, 2017; Zhang et al., 2018; Andriushchenko et al., 2023). Second, for the normalization layer, Lyle et al. (2024a) mentioned that unbounded parameters in normalization layers may cause issues in non-stationary environments such as continual or reinforcement learning. Although Summers & Dinneen (2019) demonstrated regularization for scale and shift parameters is only effective in specific situations, we also regularize scale and shift parameters, since our experiments focused on non-stationary environments and we observed that weights on several models diverged during training. Due to the different roles of regularization for each type of layer, we split the coefficient $\lambda$ into two parts in the experiments. Henceforth, we denote the coefficient for the classifier as $\lambda_c$ and the coefficient applied to the feature extractor (before the classifier) as $\lambda_f$.

### 3.4 SWR FOR IMPROVED BALANCEDNESS

One of the advantages of SWR is that it aligns the magnitude ratios between layers. Neyshabur et al. (2015a); Liu et al. (2021) have mentioned that when the balance between layers is not maintained, it has a significant negative impact on subsequent gradient descent. Although Du et al. (2018) argued

that the balance between layers is automatically adjusted during training for the ReLU network, Lyle et al. (2024b) showed that in non-stationary environments, it is common for layers to grow at different rates. Weight decay cannot resolve this issue, since when the magnitude of a specific layer increases, the regularization effect on other layers is significantly reduced (Liu et al., 2021). However, SWR, which applies regularization to each layer individually, is not affected by this issue. We will show that using SWR at every update step makes the model balanced and illustrate empirical results with a toy experiment in Appendix C.

# 4 EXPERIMENTS

In this section, we evaluate the effectiveness of SWR, comparing with other weight regularization methods. In all experiments, we used various models and datasets to compare results across different environments. For relatively smaller models, such as a 3-layer MLP and a CNN with 2 convolutional layers and 2 fully connected layers, we used MNIST (Deng, 2012), CIFAR10 and CIFAR100(Krizhevsky et al., 2009) datasets, which is commonly used in image classification experiment. To verify the effect of combining batch normalization, we additionally used a CNN-BN, which is CNN with batch normalization layers. For an extensive evaluation, we consider VGG-16 (Simonyan & Zisserman, 2014) with the TinyImageNet dataset (Le & Yang, 2015). In all the following experiments, we compared our method with two weight regularizations, L2 (Krogh & Hertz, 1991) and L2 Init (Kumar et al., 2023), as well as two re-initialization methods, Head Reset (Nikishin et al., 2022) and S&P (Ash & Adams, 2020). Detailed experimental settings, including hyperparameters for each method, are in Appendix D.

## 4.1 WARM-STARTING

We use a warm starting setup from Ash & Adams (2020) to evaluate whether SWR can close the generalization gap. In our setting, models are trained for 100 epochs with 50% of training data and trained the entire training dataset for the following 100 epochs. Re-initialization methods are applied once before the training data is updated with the new dataset.

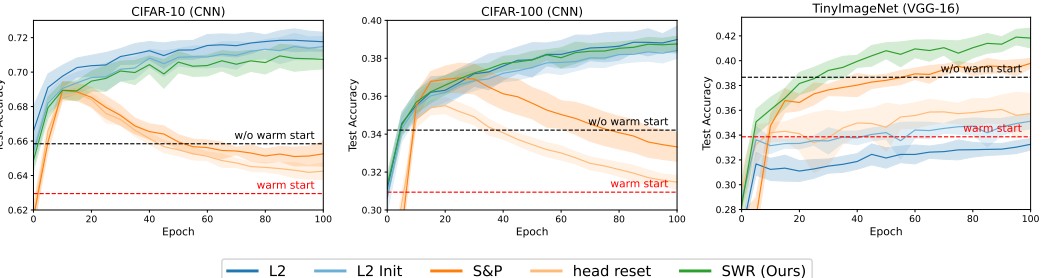

Figure 2: **Results on warm-starting.** This figure shows the test accuracy after training half of the data with 100 epochs. The dashed lines represent the final test accuracy with and without warm-start, respectively.

Fig. 2 shows the test accuracy over the 100 epochs after the dataset was added. The dashed line indicates the final accuracy of the model without applying any regularization. The red line represents the warm-start scenario, and the black line shows the model trained from scratch for 100 epochs. Weight regularization methods such as L2 regularization and L2 Init, generally exceed the accuracy of without warm-starting in most small models, but it brings no advantage for larger models like VGG-16. Re-initialization methods, S&P and resetting the last layer, perform well, occasionally surpassing the performance of models without warm-start in VGG-16. Conversely, in smaller models, they yield only marginal improvements, suggesting that using either re-initialization or regularization methods in isolation fails to fully address warm-start challenges.

However, regardless of the model size, SWR exhibited either comparable or better performance compared to other methods. In the case of VGG-16, while other regularization techniques failed to overcome the warm-start condition, SWR surpassed the test accuracy of S&P, which achieved the

highest performance among the other methods. This indicates that with proper weight regularization, models may get more advantages than with methods that reset parts of the model. We leave the additional results for the warm start in the Appendix F.

## 4.2 CONTINUAL LEARNING

In the earlier section, we examined the impact of SWR on the generalization gap and observed considerable advantages. This subsection aims to verify whether a model that is repeatedly pre-trained can continue to learn effectively. Similar to the setup provided by Shen et al. (2024), the entire data is randomly split into 10 chunks, and the training process consists of 10 stages. At each stage $k$, the model gains additional access to the $k$-th chunk. This allows us to evaluate how effectively each method can address the generalization gap when warm starts are repeated.

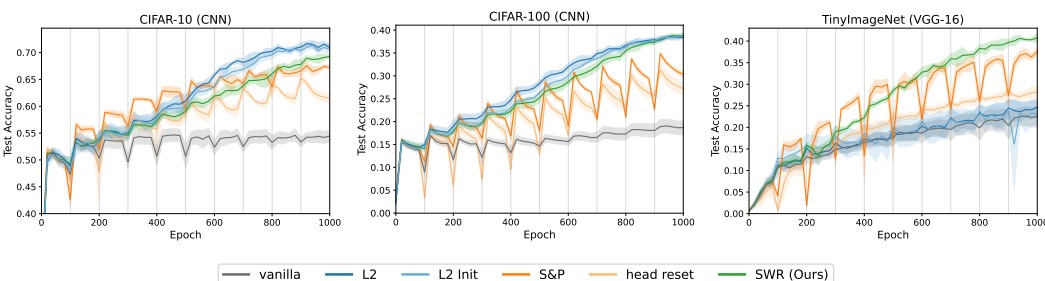

Figure 3: **Results on continual full access setting.** The test accuracy with training 10 chunks. For each chunk, the model is trained for 100 epochs and once the chunk completes training, it gets accumulated into the next chunk.

As shown in Fig. 3, the result exhibits a similar behavior as warm-start. The regularization methods steadily improve performance during the entire training process for relatively small models. The Re-init methods also achieve higher performance than the vanilla model, but it is inevitable to experience a performance drop immediately after switching chunks and applying those methods. For a larger model, VGG-16, re-initializing weights is more beneficial for learning future data than simply regularizing weights. However, from the mid-phase of training, SWR begins to outperform S&P without losing performance. It shows that re-initialization provides significant benefits in the early stages of training, it becomes evident that well-regularized weights can offer greater advantages for future performance.

Although S&P showed comparable effectiveness, such re-initialization methods lead to a loss of previously acquired knowledge. This phenomenon not only incurs additional costs for recovery but also presents critical issues when access to previous data is limited. In order to assess whether SWR can overcome these challenges, we modified the configuration; at the $k$-th stage, the model is trained only on the $k$-th chunk of data. This limited access setting restricts the model's access to previously learned data and is widely used to assess catastrophic forgetting.

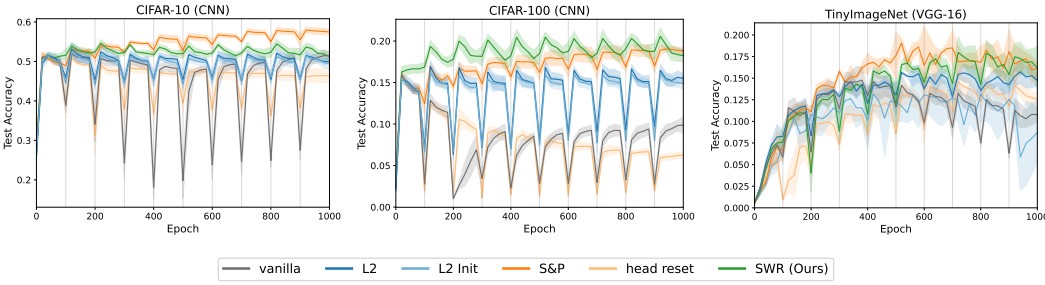

Figure 4: **Results on continual limited setting.** The test accuracy with training 10 chunks. For each chunk, the model is trained for 100 epochs and it cannot be accessed when training the next chunk.

As shown in Fig. 4, we observe that, with CNN networks, SWR loses less test accuracy than other regularization methods when the chunk of training data changes. For VGG-16, SWR maintained test accuracy without a decrease at each stage. Although at risk of losing knowledge, S&P demonstrates competitive performance with other regularization methods. This suggests that, while re-initialization and re-training can demonstrate competitive performance in some cases, the risk of losing previously acquired knowledge should not be overlooked. SWR, by contrast, mitigates this risk and maintains stability in test accuracy across stages. Further investigation is needed to explore the specific circumstances under which re-initialization may offer benefits despite the risk of information loss. Additional results for other models and datasets are provided in Appendix F.

## 4.3 GENERALIZATION

To evaluate the impact of SWR not only on plasticity but also on standard generalization performance, we conducted experiments in a standard supervised learning setting. We trained the models for a total of 200 epochs with a learning rate, 0.001. The final test accuracy is shown in Table. 1. SWR outperformed other regularization methods across most datasets and models. Notably, in larger models such as VGG-16, where other regularization techniques offered minimal performance gains, SWR achieved an improvement of over 4% in test accuracy. This indicates that more effective methods for regulating parameters exist beyond conventional techniques like weight decay, commonly employed in supervised learning.

| Method | MNIST (MLP) | CIFAR-10 (CNN) | CIFAR-100 (CNN) | CIFAR-100 (CNN-BN) | TinyImageNet (VGG-16) |
|---|---|---|---|---|---|
| vanilla | $0.9789 \pm 0.0009$ | $0.6500 \pm 0.0083$ | $0.3283 \pm 0.0067$ | $0.3234 \pm 0.0053$ | $0.3912 \pm 0.0142$ |
| L2 | $0.9795 \pm 0.0019$ | $0.7119 \pm 0.0037$ | $0.3882 \pm 0.0064$ | $\mathbf{0.4222 \pm 0.0043}$ | $0.3915 \pm 0.0108$ |
| L2 Init | $0.9793 \pm 0.0016$ | $0.7041 \pm 0.0125$ | $0.3881 \pm 0.0050$ | $0.4030 \pm 0.0105$ | $0.3870 \pm 0.0143$ |
| SWR (Ours) | $\mathbf{0.9822 \pm 0.0024}$ | $\mathbf{0.7158 \pm 0.0063}$ | $\mathbf{0.3914 \pm 0.0070}$ | $0.4129 \pm 0.0105$ | $\mathbf{0.4348 \pm 0.0025}$ |

Table 1: **Results on generalization.** The final test accuracy with training 200 epochs with a learning rate of 0.001. SWR achieves comparable or even higher performance than other simple regularization methods in stationary image classification.

To verify whether SWR works effectively with learning rate schedulers commonly used in supervised learning, we conducted additional experiments where the learning rate decays at specific epochs. Detailed results are provided in Appendix E.

## 5 CONCLUSION

In this paper, we introduced a novel method to recover the plasticity of neural networks. The proposed method, Soft Weight Rescaling, scales down the weights in proportion to the rate of weight growth. This approach prevents unbounded weight growth, a key factor behind various issues in deep learning. Through a series of experiments on standard image classification benchmarks, including warm-start and continual learning settings, SWR consistently outperformed existing weight regularization and re-initialization methods.

Our study primarily focused on scaling down parameters. However, scaling up the weights depending on the learning progress could also prove beneficial. Investigating active scaling methods could potentially address the issues associated with the extensive training time in large neural networks. Although SWR achieved impressive results in several experiments, L2 often demonstrated better performance. This suggests the potential existence of even more effective weight rescaling methods. Additionally, there are further opportunities for exploration, such as regularizing models like transformers using proportionality or investigating alternative approaches to estimating the weight growth rate. A promising approach involves analyzing initialization techniques that effectively address these challenges. This analysis could yield insights into the characteristics of model parameters, potentially leading to improved initialization or optimization methods.

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

## A    PROOF OF THEOREM 1

*Proof.* Consider a set $c = \{c_1, c_2, \ldots, c_L\}$ consisting of positive real numbers such that $C = \Pi_{i=1}^{L} c_i$. Then, construct the new parameter set $\theta^c \doteq \{W_1^c, b_1^c, \ldots W_L^c, b_L^c\}$ according to the following rules:

$$W_l^c \leftarrow c_l \cdot W_l, \quad b_l^c \leftarrow \left( \prod_{i=1}^{l} c_i \right) \cdot b_l$$

Let $a_l^c$ and $z_l^c$ denote the output after passing through the $l$-th activation function and layer, respectively. Since the homogeneous activation function $\phi$ satisfies $c\phi(x) = \phi(cx)$ for any $c \geq 0$, output of the constructed network $z^c = f_{\theta^c}(x)$ is,

$$f_{\theta^c}(x) = z^c = W_L^c a_{L-1}^c + b_L^c$$

$$= c_L \left( W_L \phi(z_{L-1}^c) + \prod_{i=1}^{L-1} c_i b_L \right)$$

$$= c_L \left( W_L \phi \left( W_{L-1}^c a_{L-2}^c + b_{L-1}^c \right) + \prod_{i=1}^{L-1} c_i b_L \right)$$

$$= c_L \left( W_L \phi \left( c_{L-1} \left( W_{L-1} \phi(z_{L-2}^c) + \prod_{i=1}^{L-2} c_i b_{L-1} \right) \right) + \prod_{i=1}^{L-1} c_i b_L \right)$$

$$= c_L c_{L-1} \left( W_L \phi \left( W_{L-1} \phi(z_{L-2}^c) + \prod_{i=1}^{L-2} c_i b_{L-1} \right) + \prod_{i=1}^{L-2} c_i b_L \right)$$

$$= \ldots$$

$$= c_L c_{L-1} \ldots c_1 \cdot f_\theta(x)$$

$$= C \cdot f_\theta(x)$$

Therefore, we can construct proportional networks with proportionality constant $C$ using infinitely many set $c$. $\square$

## B  BOUNDEDNESS

In this section, we present the proof for the weight magnitude boundedness of SWR. If the Frobenius norm of the weight of an arbitrary layer is bounded by a constant, the entire network is also bounded. Therefore, we focus on demonstrating the boundedness of a single layer.

**Theorem 2.** *If the change of squared Frobenius norm of the weight matrix, resulting from the single gradient update, is bounded by a constant for all weight matrices in the neural network, then SWR for every update step with fixed coefficient $\lambda$ bounds the Frobenius norm of the weight matrix.*

*Proof.* It is enough to show the case where the gradient update increases the magnitude of the weight matrix. For a weight matrix in step $t \geq 1$, $W_t$, let the matrix after applying SWR with $\lambda$ once be $W_t^c$, $W_{t-1}^c$ be the weight matrix before the gradient update at $W_t$, and $B > 0$ be the bound of the change of squared Frobenius norm of the matrix. $W_t^c$ can be written as below:

$$W_t^c = \frac{\lambda \times \|W_0\| + (1-\lambda) \times \|W_t\|}{\|W_t\|} W_t \tag{1}$$

$$= \left( \lambda \frac{\|W_0\|}{\|W_t\|} + (1-\lambda) \right) W_t \tag{2}$$

The reduction of the Frobenius norm by scaling can be simply represented as:

$$\|W_t\| - \|W_t^c\| = \|W_t\| - \left( \lambda \frac{\|W_0\|}{\|W_t\|} + (1-\lambda) \right) \|W_t\| \tag{3}$$

$$= \|W_t\| - (\lambda \|W_0\| + (1-\lambda)\|W_t\|) \tag{4}$$

$$= \lambda(\|W_t\| - \|W_0\|) \tag{5}$$

From the assumption, the increase of the Frobenius norm by gradient update is bounded.

$$B \geq \left| \|W_t\|^2 - \|W_{t-1}^c\|^2 \right| \tag{6}$$

$$= \left| \|W_t\| - \|W_{t-1}^c\| \right| \times \left| \|W_t\| + \|W_{t-1}^c\| \right| \tag{7}$$

$$\geq \left| \|W_t\| - \|W_{t-1}^c\| \right|^2 \tag{8}$$

$$\implies \left| \|W_t\| - \|W_{t-1}^c\| \right| \leq \sqrt{B} \tag{9}$$

From the perspective of the Frobenius norm, the weight magnitude stops growing when the reduction with scaling gets greater than the increase with gradient update. The condition can be written by below inequality:

$$\lambda(\|W_t\| - \|W_0\|) \geq \sqrt{B} \tag{10}$$

$$\|W_t\| \geq \frac{\sqrt{B}}{\lambda} + \|W_0\| \doteq B' \tag{11}$$

For all $t \geq 1$, if the Frobenius norm exceeds $B'$, it will no longer increase. Since $B'$ is constant, we can bound the Frobenius norm as follows:

$$\|W_t\| \leq B' \tag{12}$$

$\square$

By following the assumptions of Theorem 2, it can be easily shown that the weight Frobenius norm growth follows $O(\sqrt{t})$ as the empirical evidence shown in (Merrill et al., 2020), thereby indicating that the assumption is not unreasonable.

Since the spectral norm of the weight matrix is lower than its Frobenius norm, we can show that the neural network using SWR has an upper bound of the Lipschitz constant. For simplexity, we only consider MLP with a 1-Lipschitz activation function.

**Corollary 2.1.** *For an MLP, $f_\theta$, with $1$-Lipshcitz activation function (e.g. ReLU, Leaky ReLU, etc.), $f_\theta$ is Lipschitz continuous with applying SWR for every update step.*

*Proof.* We denote the spectral norm of the matrix with $\|\cdot\|_\sigma$. Let weight matrices of $f_\theta$ be $W^l$ ($l \in \{1, 2, \ldots L\}$), and $B^l$ be the upper bound of the Frobenius norm of each of them. Using the relationship between the Frobenius norm and the spectral norm, $\|W^l\|_\sigma \leq \|W^l\|$ for all $l$. Since the Lipschitz constant of the weight matrix is same with its spectral norm and composition of $l_1$ and $l_2$ Lipschitz function is $l_1 l_2$ Lipschitz function (Gouk et al. (2021)), the Lipschitz constant of neural network $k_\theta$ can be express as:

$$k_\theta \leq \prod_l \|W^l\|_\sigma \tag{13}$$

$$\leq \prod_l \|W^l\| \tag{14}$$

$$\leq \prod_l B^l \doteq B' \tag{15}$$

Note that the Lipschitz constant of the activation function is 1, so activation functions do not affect to bound of the Lipschitz constant of $k_\theta$. Since Lipschitz constant $k_\theta$ is bounded with $B'$, $f_\theta$ is $B'$-Lipschitz continuous function. $\square$

Similarly, we can get the neural network that is trained with SWR as Lipschitz continuous when using a convolution network or normalization layer. We left a tight upper bound of Lipschitz constant for future work.

## C BALANCEDNESS

### C.1 EMPIRICAL STUDY

Neyshabur et al. (2015a) defined the entry-wise $\ell_{p,q}$-norm of the model, which is expressed as follows:

$$\|W\|_{p,q} = \left( \sum_i \left( \sum_j |W_{ij}|^p \right)^{\frac{q}{p}} \right)^{\frac{1}{q}}. \tag{16}$$

If two models are functionally identical, the model that has a smaller $\ell_{p,q}$-norm represents more balanced. In order to estimate the model balancedness, we used the ratio between the entry-wise $\ell_{p,q}$-norm of current and global minimal. We compute the global minimal $\ell_{p,q}$-norm using Algorithm 1 of Saul (2023). Fig. 5 shows the balancedness of the 3-layer MLP, measured at the end of each epoch, along with the test accuracy. SWR is shown to enhance model balancedness and improve test accuracy compared to the vanilla model.

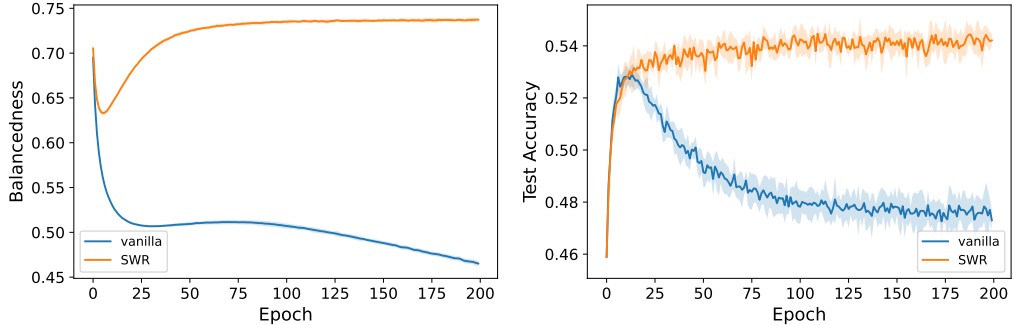

Figure 5: **Results for the balancedness.** The left figure shows the balancedness of the model, and the right figure shows the test accuracy. The results are averaged over 5 runs on CIFAR-10 dataset.

## C.2 THEORETICAL ANALYSIS

Next, we will show that SWR improves the balance between layers. Before proving it, we define how to express balancedness.

**Definition 2** (Balancedness between two layers). *Consider a network with two weight matrices at time step $t$ to be $W_t$ and $W'_t$ (at initial, $W_0, W'_0$). Without loss of generality, we let $\|W_0\| \leq \|W'_0\|$. We define the balance of two layers $b_t$ as the difference of rates of the Frobenius norms of weight matrices from the initial state. This can be expressed as follows:*

$$b_t \doteq |r_t - r_0|, \text{ where } r_t = \frac{\|W'_t\|}{\|W_t\|} \tag{17}$$

That is, $b_t$ is a non-negative value, and the closer it is to 0, the better balance between the two layers.

**Theorem 3.** *Applying SWR with coefficient $\lambda$ enhances the balance of the neural network.*

*Proof.* Keep the settings from Definition 2. Let $W_t$ and $W'_t$ be the weight matrices of any two layers at time step $t$ in the neural network and $b_t$ be the balance of $W_t$ and $W'_t$. Then, $b^c_t$, the balance after applying SWR with coefficient $\lambda$, can represent it as below:

$$b^c_t \doteq |r^c_t - r_0|, \text{ where } r^c_t = \frac{\|W'^c_t\|}{\|W^c_t\|} \tag{18}$$

where $W^c_t$ and $W'^c_t$ are the weight matrices that scaled by SWR with $\lambda$. Then, by equation 5, $r^c_t$ can be expanded as follows:

$$r^c_t = \frac{\lambda\|W'_0\| + (1-\lambda)\|W'_t\|}{\lambda\|W_0\| + (1-\lambda)\|W_t\|} \tag{19}$$

Since $r^c_t$ is the form of generalized mediant of $r_t$ and $r_0$, if $r_0 \leq r_t$, the relationship between their magnitudes and balance satisfies as below:

$$
\begin{align}
r_0 &\leq r^c_t &\leq r_t \tag{20} \\
\Rightarrow \quad 0 &\leq r^c_t - r_0 &\leq r_t - r_0 \tag{21} \\
\Rightarrow \quad 0 &\leq |r^c_t - r_0| &\leq |r_t - r_0| \tag{22} \\
\Rightarrow \quad 0 &\leq b^c_t &\leq b_t \tag{23}
\end{align}
$$

If $r_0 \geq r_t$, we can derive equation 22, following a similar approach. Therefore, the balance of arbitrary two layers gets better when applying SWR, which indicates an overall improvement in the balance across all layers of the network. $\square$

# D   DETAILS FOR EXPERIMENTAL SETUP

In this section, we will provide details on the experimental setup. First, we specify the hyperparameters that we commonly use. We used 256 for the batch size of the mini-batch and 0.001 for the learning rate. The Adam optimizer was employed, with its hyperparameters set to the default values without any modification. We employed distinct 5 random seeds for all experiments while performing 3 seeds for VGG-16 due to computational efficiency. In the following sections, we present model architectures, the baseline methods that we compared, and the hyperparameters for the best test accuracy.

## D.1   MODEL ARCHITECTURES

We utilized four model architectures consistently throughout all experiments. The detailed information on architectures is as follows:

**MLP**: We used the 3-layer Multilayer Perceptron (MLP) with 100 hidden units. The 784 ($28 \times 28$) input size and 10 output size are fixed since MLP is only trained in the MNIST dataset.

**CNN**: We employed a Convolutional Neural Network (CNN), which is used in relatively small image classification. The model includes two convolutional layers with a $5 \times 5$ kernel and 16 channels. The fully connected layers follow with 100 hidden units.

**CNN-BN**: In order to verify whether our methodology is effectively applied to normalization layers, we attached batch normalization layers following the convolutional layer in the CNN model.

**VGG-16** (Simonyan & Zisserman, 2014): We adopted VGG-16 to investigate whether SWR adapts properly in large-size models. The number of hidden units of the classifiers was set to 4096 without dropout.

## D.2   BASELINES

**L2.** The L2 regularization is known as enhancing not only generalization performance Krogh & Hertz (1991) but also plasticity Lyle et al. (2024b). We add the loss term $\frac{\lambda}{2}\|\theta\|^2$ on the cross-entropy loss. We swept $\lambda$ in $\{0.1, 0.01, 0.001, 0.0001, 0.00001\}$.

**L2 Init.** Kumar et al. (2023) introduced a regularization method to resolve the problem of the loss of plasticity where the input or output of the training data changes periodically. They argued that regularizing toward the initial parameters, results in resetting low utility units and preventing weight rank collapse. We add the loss term $\frac{\lambda}{2}\|\theta - \theta_0\|^2$ on the cross-entropy loss, where $\theta_0$ is the initial learnable parameter. We performed the same grid search with L2.

**S&P.** Ash & Adams (2020) demonstrated that the network loses generalization ability for warm start setup, and introduced effective methods that shrink the parameters and add noise perturbation, periodically. In order to reduce the complexity of hyperparameters, we employ a simplified version of S&P using a single hyperparameter, as shown in Lee et al. (2024). We applied S&P when the training data was updated. The mathematical expression is $\theta \leftarrow (1 - \lambda)\theta + \lambda\theta_0$, where $\theta_0$ is initial learnable parameters, and we swept $\lambda$ in $\{0.2, 0.4, 0.6, 0.8\}$.

**head reset.** Nikishin et al. (2022) suggested that periodically resetting the final few layers is effective in mitigating plasticity loss. In this paper, we reinitialized the fully connected layers with the same period with S&P. We only applied reset to the final layer, when MLP is used for training.

**SWR.** For networks that do not have batch normalization layers, we swept $\lambda$ in $\{1, 0.1, 0.01, 0.001, 0.0001\}$. Otherwise, we performed a grid search for $\lambda_c$ and $\lambda_f$ in the same range of $\lambda$.

Table. 2-4 shows the best hyperparameter set that we found in various experiments.

# E   GENERALIZATION RESULTS WITH LEARNING RATE DECAY

To assess the performance of SWR under the learning rate scheduler, we conducted learning rate decay in Experiment 4.3. The rest of the configuration was kept unchanged, while the learning rate was multiplied by 1/10 at the start of the 100th and 150th epochs.

| Dataset | Method | Hyperparameter Set |
|---|---|---|
| MNIST (MLP) | S&P | $\lambda = 0.4$ |
| | L2 | $\lambda = 1e{-}5$ |
| | L2 Init | $\lambda = 1e{-}5$ |
| | SWR | $\lambda = 1e{-}4$ |
| CIFAR-10 (CNN) | S&P | $\lambda = 0.8$ |
| | L2 | $\lambda = 1e{-}2$ |
| | L2 Init | $\lambda = 1e{-}2$ |
| | SWR | $\lambda = 1e{-}3$ |
| CIFAR-100 (CNN) | S&P | $\lambda = 0.8$ |
| | L2 | $\lambda = 1e{-}2$ |
| | L2 Init | $\lambda = 1e{-}2$ |
| | SWR | $\lambda = 1e{-}3$ |
| CIFAR-100 (CNN-BN) | S&P | $\lambda = 0.8$ |
| | L2 | $\lambda = 1e{-}2$ |
| | L2 Init | $\lambda = 1e{-}2$ |
| | SWR | $\lambda_f = 1e{-}4, \lambda_c = 1e{+}0$ |
| TinyImageNet (VGG-16) | S&P | $\lambda = 0.8$ |
| | L2 | $\lambda = 1e{-}5$ |
| | L2 Init | $\lambda = 1e{-}5$ |
| | SWR | $\lambda_f = 1e{-}2, \lambda_c = 1e{-}1$ |

Table 2: Hyperparameter set of each method on the warm-start experiment.

| Dataset | Method | Full Access | Limited Access |
|---|---|---|---|
| MNIST (MLP) | S&P | $\lambda = 0.6$ | $\lambda = 0.2$ |
| | L2 | $\lambda = 1e{-}4$ | $\lambda = 1e{-}5$ |
| | L2 Init | $\lambda = 1e{-}4$ | $\lambda = 1e{-}5$ |
| | SWR | $\lambda = 1e{-}4$ | $\lambda = 1e{-}4$ |
| CIFAR-10 (CNN) | S&P | $\lambda = 0.8$ | $\lambda = 0.4$ |
| | L2 | $\lambda = 1e{-}2$ | $\lambda = 1e{-}2$ |
| | L2 Init | $\lambda = 1e{-}2$ | $\lambda = 1e{-}2$ |
| | SWR | $\lambda = 1e{-}3$ | $\lambda = 1e{-}1$ |
| CIFAR-100 (CNN) | S&P | $\lambda = 0.8$ | $\lambda = 0.6$ |
| | L2 | $\lambda = 1e{-}2$ | $\lambda = 1e{-}2$ |
| | L2 Init | $\lambda = 1e{-}2$ | $\lambda = 1e{-}2$ |
| | SWR | $\lambda = 1e{-}3$ | $\lambda = 1e{-}1$ |
| CIFAR-100 (CNN-BN) | S&P | $\lambda = 0.8$ | $\lambda = 0.4$ |
| | L2 | $\lambda = 1e{-}2$ | $\lambda = 1e{-}2$ |
| | L2 Init | $\lambda = 1e{-}2$ | $\lambda = 1e{-}2$ |
| | SWR | $\lambda_f = 1e{-}4, \lambda_c = 1e{-}1$ | $\lambda_f = 1e{-}1, \lambda_c = 1e{-}2$ |
| TinyImageNet (VGG-16) | S&P | $\lambda = 0.8$ | $\lambda = 0.4$ |
| | L2 | $\lambda = 1e{-}4$ | $\lambda = 1e{-}4$ |
| | L2 Init | $\lambda = 1e{-}4$ | $\lambda = 1e{-}3$ |
| | SWR | $\lambda_f = 1e{-}2, \lambda_c = 1e{-}2$ | $\lambda_f = 1e{-}4, \lambda_c = 1e{+}0$ |

Table 3: Hyperparameter set of each method on continual learning.

There is a consideration to be addressed when applying learning rate decay with SWR. When the learning rate decays, we will show that the regularization strength that maintains balance becomes relatively stronger. Suppose that after time step $t$, the L2 norm of the weight vector is near convergence. To simplify the case, let us assume the weight vector, $w_t$, aligns with the direction of the gradient of the loss $\nabla_w L(w)$. After the SGD update, the weight vector will be updated as $w_{t+1} = w_t - \alpha \nabla_w L(w)$, meaning the change of L2 norm is $\alpha \|\nabla_w L(w)\|$.

According to equation 5, when applying SWR, the change in L2 norm becomes $\lambda(\|w_{t+1}\| - \|w_0\|)$. Under our assumption, we have $\alpha \|\nabla_w L(w)\| \approx \lambda(\|w_{t+1}\| - \|w_0\|)$. Therefore, when a learning rate decay occurs, this equivalence is broken, causing the weight norm to drop toward the initial weight norm. To address this issue, we used a simple trick that reset the initial weight norm to the current norm when decay happens, as $n^{\text{init}} \leftarrow \|w_t\|$. We refer to this method as SWR + re-init.

The results with learning rate decay can be found in Table 5. SWR+re-init demonstrated performance largely comparable to other methods, specifically leading to an improvement of over 8% in

| Dataset | Method | Hyperparameter Set |
|---|---|---|
| MNIST (MLP) | L2 | $\lambda = 1e-5$ |
| | L2 Init | $\lambda = 1e-5$ |
| | SWR | $\lambda = 1e-4$ |
| CIFAR-10 (CNN) | L2 | $\lambda = 1e-2$ |
| | L2 Init | $\lambda = 1e-2$ |
| | SWR | $\lambda = 1e-3$ |
| CIFAR-100 (CNN) | L2 | $\lambda = 1e-2$ |
| | L2 Init | $\lambda = 1e-2$ |
| | SWR | $\lambda = 1e-3$ |
| CIFAR-100 (CNN-BN) | L2 | $\lambda = 1e-2$ |
| | L2 Init | $\lambda = 1e-2$ |
| | SWR | $\lambda_f = 1e-4, \lambda_c = 1e-1$ |
| TinyImageNet (VGG-16) | L2 | $\lambda = 1e-5$ |
| | L2 Init | $\lambda = 1e-5$ |
| | SWR | $\lambda_f = 1e-2, \lambda_c = 1e-1$ |

Table 4: Hyperparameter set of each method on generalization experiment.

test accuracy on VGG-16. While SWR + re-init generally outperformed standalone SWR, a slight performance drop was observed in larger models such as VGG-16. This suggests that more effective solutions exist to handle this issue when using learning rate decay. Further research on this matter will be left as future work.

| Method | MNIST (MLP) | CIFAR-10 (CNN) | CIFAR-100 (CNN) | CIFAR-100 (CNN-BN) | TinyImageNet (VGG-16) |
|---|---|---|---|---|---|
| vanilla | $0.9798 \pm 0.0005$ | $0.6571 \pm 0.0057$ | $0.3490 \pm 0.0021$ | $0.3483 \pm 0.0043$ | $0.4126 \pm 0.0236$ |
| L2 | $0.9811 \pm 0.0007$ | $\mathbf{0.7304 \pm 0.0039}$ | $0.3949 \pm 0.0091$ | $\mathbf{0.4532 \pm 0.0040}$ | $0.4080 \pm 0.0124$ |
| L2 Init | $0.9811 \pm 0.0007$ | $0.7286 \pm 0.0023$ | $0.4048 \pm 0.0019$ | $0.4341 \pm 0.0019$ | $0.4199 \pm 0.0048$ |
| SWR (Ours) | $0.9796 \pm 0.0009$ | $0.6925 \pm 0.0078$ | $0.3599 \pm 0.0054$ | $0.4240 \pm 0.0015$ | $\mathbf{0.5221 \pm 0.0123}$ |
| SWR + re-init (Ours) | $\mathbf{0.9829 \pm 0.0002}$ | $0.7269 \pm 0.0027$ | $\mathbf{0.4133 \pm 0.0058}$ | $0.4451 \pm 0.0028$ | $0.5165 \pm 0.0070$ |

Table 5: **Results on generalization with learning rate decay.** The final test accuracy after training 200 epochs. The learning rate initialized with 0.001 and divided by 10 at epoch 100 and 150.

# F    ADDITIONAL RESULTS

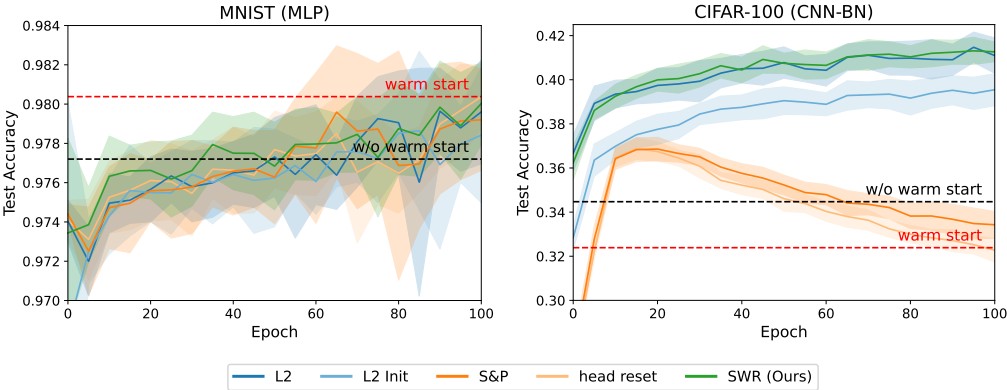

Figure 6: **Additional results on warm-starting.**

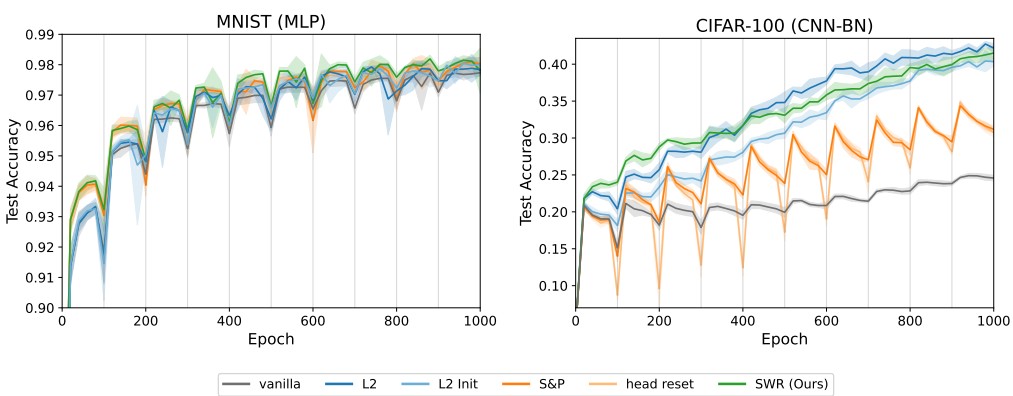

Figure 7: **Additional results on continual full access setting.**

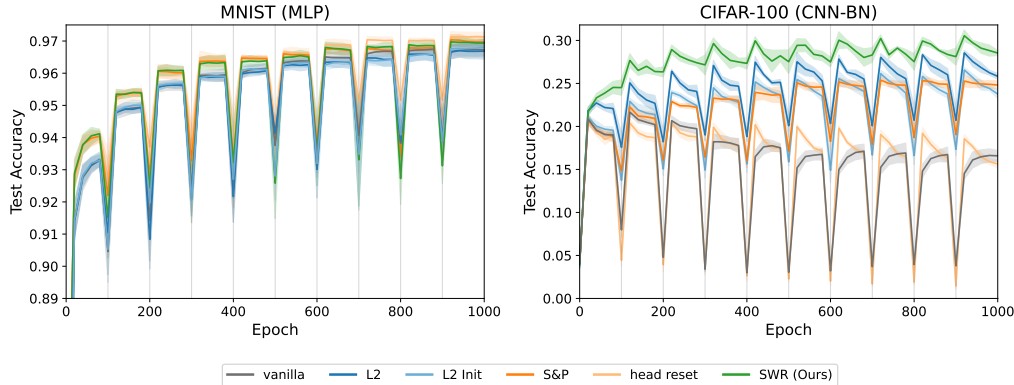

Figure 8: **Additional results on continual limited access setting.**

