# OpenReview forum: "Recovering Plasticity of Neural Networks via Soft Weight Rescaling"
_ICLR.cc/2025/Conference — Submitted to ICLR 2025_

### Official Review · Reviewer_CTTf · 2024-10-28

**Soundness:** 2
**Presentation:** 1
**Contribution:** 2
**Rating:** 3
**Confidence:** 4

**Summary:**

This paper introduces Soft Weight Regularization (SWR), a regularization based algorithm for maintaining plasticity under the broad framework of continual learning. Unlike other regularization based approaches for addressing plasticity loss, such as L2 regularization, Shrink and Perturb, and L2 Init, SWR does not alter the network's predictions. The paper provides a theoretical analysis showing that SWR bounds weight magnitudes and maintains balanced weights between layers, two favourable properties of neural networks. Finally, the paper provides empirical evidence arguing the efficacy of SWR on a set of problems that test for plasticity and stability in settings of warm-starting, continual learning, and generalization.

**Strengths:**

- The paper rightfully makes the point that unbounded weight magnitudes in continual learning settings is a more general issue in deep learning. This is a point that is not often made explicit in the continual learning literature.
- The proposed method of SWR is supported by theoretical analysis establishing that weight magnitudes are bounded and that weights between layers are relatively balanced, two properties that have been previously shown to be beneficial for generalization and continual learning settings. Many recent methods in the continual learning framework, despite their simplicity, have been introduced with little to no theoretical basis, therefore, this is  a strength of this paper.
- The proposed method is evaluated on three types of problems: warm-starting, continual learning, and classic supervised learning evaluating generalization. SWR's performance is evaluated with respect to the generalization gap, plasticity in continual learning, and catastrophic forgetting in continual learning. This provides a broader evaluation than is typical in continual learning.

**Weaknesses:**

- This paper could use more polish and could be reorganized to better state the contributions as well as their relative merits to existing work. Some concrete examples are as follows:
- The paragraph on line 039 is too specific for the introduction and the paper would be better served with a concise overview of the merits and draw backs of regularization based re-initialization based methods, and moving the existing paragraph as is to a related works section.
- The paragraph on line 066 is redundant given the preceding paragraph.
- It would be useful to give, at least a high level or rough, description of SWR in the introduction so that the reader has an understanding of how SWR differs from existing regularization based methods. As the paper is currently written, SWR is described by its merits: bounded weight magnitudes and balancing weights, and no actual description of the algorithm itself is provided, until the full algorithm is presented on page 5.
- It would be useful to introduce and define both catastrophic forgetting and plasticity in the introduction, rather than just the latter phenomenon, as the paper claims to evaluate SWR's ability to mitigate catastrophic forgetting.
- The motivating and illustrating experiment, Figure 1 and the paragraph that follows on line 197 are confusing and I cannot make out the experimental setup and the exact point that is being made. I would suggest explicitly describing the experimental setup and each algorithm that you are evaluating. How exactly are you scaling, and what is scaling with and without proportionality in this example? Does the pre-trained model include any scaling? What is the difference between fine-tuned after scaling and just the scaled model? When you train the fine-tuned model for another 50 epochs, are you fine tuning on the validation set or some new training set? What exactly is the scaling magnitude or scaling ratio in this experiment? Given that this is a motivating or illustrating example, it would be useful to very precise with outlining the experimental setup.
- I would recommend moving your theorems on boundedness and balancedness to section 3 and commenting on the significance of these theorems rather than pointing the reader to the appendix.
- The set of competitor algorithms is limited. Specifically, for re-initialization based methods the well-cited Continual Backprop (Dohare et al.) and ReDO (Sokar et al.) are missing from the experiments that evaluate plasticity loss. As for the experiment that evaluates catastrophic forgetting, regularization based methods for explicitly addressing this phenomenon such as Elastic Weight Consolidation (Kirkpatrick et al.) are absent.
- To evaluate the efficacy of SWR for mitigating plasticity loss and catastrophic forgetting, a wider experimental study may be necessary. You could consider the benchmark problems of Permuted MNIST, Random Label MNIST and CIFAR, and Continual ImageNet, which are nicely described in (Kumar et al).
- The claim that SWR mitigates catastrophic forgetting requires more evidence than a single experiment, as noted in the previous point. SWR does not modify the network's outputs unlike other regularization based methods, but this does not prove that SWR mitigates plasticity loss. There is a series of regularization based methods, e.g. Elastic Weight Consolidation, that regularize networks towards weights (or equivalently representations) learned during earlier tasks, and in turn mitigating catastrophic forgetting. Therefore, the limited experiments and construction of SWR do not provide sufficient evidence that catastrophic forgetting is alleviated by SWR more efficiently than by other algorithms, therefore the claims that SWR maintains useful information while re-initialization based methods do not, is not entirely accurate.

**Questions:**

- How sensitive is SWR to its choice of hyperparameter(s), it would be nice to see these results.
- Is there a reason why Theorem 2, Corollary 2.1, and Theorem 3 are not presented in the main body of the paper?
- Could you elaborate on why Shrink and Perturb experiences declining performance on the warm-starting experiments (CIFAR-10 and CIFAR-100), even though Ash and Adams introduce Shrink and Perturb and show that it is performant on these sorts of experiments?
- Can you restate Figure 1 and its experimental setup clearly, as described in the weaknesses section.

---

### Official Review · Reviewer_QFvH · 2024-11-02

**Soundness:** 3
**Presentation:** 3
**Contribution:** 3
**Rating:** 3
**Confidence:** 4

**Summary:**

The paper focuses on the solution to recovering the plasticity of DNNs via weight regularization. The paper proposes a simple yet effective weight regularization method that prevents unbounded weight growth. The authors also provided the technique's theoretical and empirical insights, which prove the generalization performance in different learning.

**Strengths:**

- This work progressively establishes and justifies its framework, making this paper easy to follow.
- The results are promising, however, I have some concerns regarding the results as discussed below

**Weaknesses:**

- One main drawback of the paper is the limited application of the paper. The authors made many assumptions (e.g., the network is affine, homogeneous with ReLU), which impedes the contributions and the applicability of the paper in real-world scenarios.
- Some crucial statements are made without proper references. Furthermore, these statements are conflicted with the statements in various peer-reviewed and significant publications.
- The paper came up with many theorems and definitions without explaining the usages and necessities of these statements.
- Ablation tests according to Theorem 1 needed to be conducted to verify the paper's significance.
- All in all, the aforementioned issue impedes the contribution and significance of the paper method. The authors please consider carefully about these issues. If the issues are addressed, the score can be modified.
- The experimental evaluations are not sufficient, they need to provide more experiments on large-scale datasets (ImageNet1K, COCO, etc) and across different model architectures (VisionTransformers, etc).
- The hyper-parameter $\lambda$ is proposed but there are no experiments that consider the effect of $\lambda$ on the boundedness of the weight before and after scaling.
- There should be a theoretical discussion about how to tighten the boundedness compared to other methods. For example, in Theorem 2, the authors show that $\|W_t\| \neq B$, which is trivial thus not proving that the proposed method is better than others.

**Questions:**

1. Can you discuss more the statement in L086: "weight growth is inevitable in deep learning"? We agree that a large value of weight norm impedes the model generalization. However, this phenomenon is usually at the initial phase of DL. It can be proved via empirical experiments [R1], or theoretical [R2, Theorem 1].
2. Moreover, in the cited paper, the authors mentioned the phenomenon when the weight norm is large. However, the authors did not mention that the weight norm is high due to the progress of the training model (which is related to the plasticity effects of the pre-trained model, which is already trained). Please note that this statement is crucial to assess the paper's contribution and significance.

[R1] Yang You, Jing Li, Sashank Reddi, Jonathan Hseu, Sanjiv Kumar, Srinadh Bhojanapalli, Xiaodan Song, James Demmel, Kurt Keutzer, Cho-Jui Hsieh, Large Batch Optimization for Deep Learning: Training BERT in 76 minutes, ICLR 2020.

[R2] Jianyu Wang, Qinghua Liu, Hao Liang, Gauri Joshi, H. Vincent Poor, Tackling the Objective Inconsistency Problem in Heterogeneous Federated Optimization, NIPS 2020.

3. What is the necessity of the notations of the network layer defined as in L130-136?
4. In 156, $f_{\theta^\prime}(x) = k\cdot f_{\theta}$ is the proportion of the output of the model, is it applicable to models with various types of activation functions or only applicable to linear activation functions?
5. What is the meaning of Theorem 1? Why do we need to find many networks that are proportional with $f_{\theta}$?
6. In L216 - L218, can the authors discuss more the statement that the "initial weight norm is small in most initialization"? To be frank, this statement needed to be considered carefully (e.g., making ablation tests or empirical evaluations).
7. In L245, why $C$ is set to 1? Is it different in performance if we set $C$ to different values? An ablation test according to the difference initial C should be made to verify the paper's method.

---

### Official Review · Reviewer_eg35 · 2024-11-04

**Soundness:** 3
**Presentation:** 3
**Contribution:** 2
**Rating:** 5
**Confidence:** 3

**Summary:**

The authors introduce Soft Weight Rescaling (SWR), a novel weight regularization method that prevents unbounded weight growth to preserve information and maintain network plasticity. The theoretical analysis shows that SWR bounds weight magnitudes and balances them across layers without degrading model performance. Empirical evaluations, particularly with VGG-16, show that SWR improves generalization performance compared to other regularization methods.

**Strengths:**

- The paper is overall clearly written and the method is adequately described.
- The proposed method SWR is computationally more efficient than previously proposed methods.
- The experiment results and analysis provided in the paper are insightful.

**Weaknesses:**

- The experimental results on smaller models are quite weak. For example, in warm-start and continual learning experiments, L2 (or S&P) seems to be better in most experiments (including the ones in the appendix). Even in Table 1, except for VGG, I wouldn't say the improvements are significantly higher since there's quite a bit of overlap with L2 in terms of standard deviations in MLP, and CNN cases. SWR only performs well on VGG which is not a very popular architecture even for vision-based experiments in this domain compared to ResNet. It would be interesting to see the comparison between SWR and baselines on bigger models. The assumptions of affine, conv layers in Theorem 1 are also strong and limit the applicability of SWR.
- I think the main novelty of the idea is limited and comes primarily from "scaling the bias vectors according to a certain rule". From Eq on line 220, one may assume that $W_l$ will attain a higher magnitude than $W_{init}$. As a result, $c_l \approx 1 - \lambda$, which implies that SWR would behave like a layer-wise version of S&P with weight_scale = $1 - \lambda$ and no initial weights.
- Missing baselines: Lyle et al. 2024 recently also showed that the L2 + Layer norm generally outperforms the majority of the existing methods. Lee et al. 2024 have also shown that their method results in superior generalization performance on these benchmarks.



Some grammatical/clarity related issues:
- Line 161: investigated the following theorem shows that
- Line 213: the change rate

**Questions:**

There are some claims made in the paper that require evidence/clarification:
- While the proposed method is computationally more efficient, it is also true that the overhead cost of regularization methods like L2 is *not* significantly high as claimed in the paper unless higher-order computation is involved. In fact, L2 is quite common even in large-scale models. Some methods only involve computing scores based on the layer outputs which is not *very* expensive.
The computational cost is only significant if there is higher-order computation involved.
- Line 382-386: We don't entirely lose previous knowledge in S&P. Rather, adding noise ultimately helps in better generalization. Even in the case of Lee et al. 2024 paper, they showed better generalization for a re-initialization method which is crucial.

---

### Official Review · Reviewer_quxT · 2024-11-04

**Soundness:** 2
**Presentation:** 2
**Contribution:** 1
**Rating:** 5
**Confidence:** 3

**Summary:**

The paper addresses the issue of plasticity loss in neural networks, where the capacity to learn new information diminishes over time due to unbounded weight growth. The authors propose a method called Soft Weight Rescaling (SWR), which mitigates this issue by scaling down the weights at each learning step, and claiming to maintain the network's plasticity without losing previously learned information. Some experimental results, such as continual leaning and single-task in image classification, demonstrate that SWR can enhance performance, outperforming existing weight regularization and re-initialization techniques.

**Strengths:**

1. The paper is easy to follow.
2. I think the authors are focusing on an interesting topic, i.e. loss of plasticity, that is worthy to probe.
3. The method proposed is simple and can be easily implemented in practice.

**Weaknesses:**

1. An unbounded weight growth is one of the main causes of plasticity loss, and the authors propose reducing weight magnitude through weight scaling. Reducing the weight magnitude could be a common implementation in training, where L2 is widely used. So I think the key here lies in comparing the proposed method to L2. However, after reviewing the text, I did not find a clear rationale why we should choose the proposed method over L2. Could the authors provide specific cases that demonstrate the essence regarding how the proposed method targets improvements over L2 regularization?

2. I notice that the authors define the rate of how much the model has changed from the initial state as the ratio between the Frobenius norm of the current weight matrix and that of the initial one. Could the author give more explanations regarding this metric? In my opinion, this metric may not well capture the extent of change in the model. For instance, applying weight regularization could significantly alter the weights, yet the model's performance may change only marginally.

3. I have not found any theoretical insights regarding the claims made about magnitude boundedness and weight balance in the main text. However, I did locate some proofs in the appendix. Since these proofs appear to be one of the main contributions of the proposed work, I recommend that the authors reorganize the paper to better highlight this important content.

4. I think the authors should improve the experiments presented in the paper. Firstly, the current training performance falls significantly below existing baselines, with VGG achieving only 0.72 on CIFAR-10 and below 0.4 on CIFAR-100, which is unacceptable. Secondly, the authors should broaden their experimental scope beyond VGG on CIFAR, MNIST, and TinyImage. It would be beneficial to include experiments relevant to current RL or NLP scenarios, especially where pre-trained models are commonly utilized. For now, I could barely sense the superiority of the proposed method.

5. It would be helpful if the authors could release the code.

**Questions:**

See Weakness.

**Details Of Ethics Concerns:**

I have not found any discussions about the limitations and potential negative societal impact. But in my opinion, this may not be a problem, since the work only focuses on the optimization in deep learning. Still, it is highly encouraged to add corresponding discussions.

---

### Meta-Review · Area_Chair_wqCH · 2024-12-17

**Metareview:**

The paper addresses the issue of plasticity loss, also known as intransigence, in neural networks. The authors identify unbounded weight growth as a key contributor to this issue and introduce a novel regularization technique called Soft Weight Rescaling (SWR) to overcome it. SWR aims to limit weight magnitudes and ensure balance across different layers without compromising model performance. The authors offer a theoretical analysis to confirm that SWR effectively maintains bounded and balanced weights, which are desirable properties in neural networks. The empirical results validate the effectiveness of SWR in various scenarios, including warm-starting, continual learning, and generalization, demonstrating its ability to preserve both plasticity and stability.

**Strengths:** All reviewers agreed that the paper addresses an interesting and timely problem. They praised the clear and methodical writing style and the method's simplicity. Moreover, the reviewers found the experimental results and analysis presented in the paper to support the claims and provide insightful observations effectively.

**Weaknesses:** The paper was primarily criticized for its limited experimental setup, which restricts its broader impact. Notably, the datasets employed, such as CIFAR10, CIFAR100, MNIST, and TinyImageNet, are small-scale. Furthermore, the use of a VGG model in the experiments resulted in performance metrics—72% on CIFAR10 and 40% on CIFAR100—that are significantly lower than current state-of-the-art results on these datasets. This discrepancy places the paper at a substantial disadvantage when compared to more recent literature. Additionally, the reviewers highlighted the absence of comparisons with related regularization methods, such as L2 regularization (weight decay) or more contemporary approaches that combine L2 and Layer Normalization (Lyle et al., 2024). Lastly, the paper lacks comprehensive ablation studies and a detailed sensitivity analysis of hyperparameters.

Although the reviewers appreciated certain aspects of the paper, they unanimously agreed that the experimental setup was rudimentary and insufficient. Consequently, they concluded that the paper, in its current form, is not ready for publication. I enjoyed reading the paper and believe that the authors could significantly enhance their work based on the constructive feedback provided by the reviewers. Considering these factors, I recommend rejecting this paper. However, I recognize its potential and encourage the authors to continue refining their work.

**Additional Comments On Reviewer Discussion:**

Unfortunately, the authors did not respond to the points raised by the reviewers. Given the consensus among reviewers on their feedback, there was little need for further discussion during the review period.

---

### Decision · Program_Chairs · 2025-01-22

Reject